# Inflammatory Bowel Disease (IBD)-Associated Colorectal Cancer (CRC): Is cGAS-STING Pathway Targeting the Key to Chemoprevention?

**DOI:** 10.3390/ijms26114979

**Published:** 2025-05-22

**Authors:** Stavros P. Papadakos, Chara Georgiadou, Alexandra Argyrou, Elisavet Michailidou, Charalampos Thanos, Stamatina Vogli, Spyros I. Siakavellas, Spillios Manolakopoulos, Stamatios Theocharis

**Affiliations:** 1First Department of Pathology, School of Medicine, National and Kapodistrian University of Athens, 11527 Athens, Greece; stpap@med.uoa.gr; 21st Department of Gastroenterology, Medical School of National and Kapodistrian University of Athens, General Hospital of Athens “Laiko”, 11527 Athens, Greece; argyalex89@gmail.com (A.A.); elisabet823@hotmail.com (E.M.); 3Department of Physiology, Medical School of National and Kapodistrian University of Athens, 75 Mikras Asias Str, 11527 Athens, Greece; chgeorgiadou@med.uoa.gr; 4Department of Gastroenterology, General Hospital of Athens “G. Gennimatas”, 11527 Athens, Greece; charalampos.a.thanos@gmail.com; 5Department of Gastroenterology, Metaxa Cancer Hospital of Piraeus, 18537 Piraeus, Greece; stamvog95@gmail.com; 6Liver-GI Unit, 2nd Academic Department of Internal Medicine, General Hospital of Athens “Hippocration”, National and Kapodistrian University of Athens, 11527 Athens, Greece; s.siakavellas@gmail.com

**Keywords:** inflammatory bowel disease (IBD), colitis-associated colorectal cancer (CAC), cGAS-STING pathway, innate immunity, chemoprevention

## Abstract

Inflammatory bowel disease (IBD)-associated colorectal cancer (CRC) remains a significant clinical challenge due to its link with chronic inflammation and the inherent limitations of current prevention and surveillance strategies. The cGAS-STING pathway has emerged as a key player in the immune regulation of inflammation-driven carcinogenesis, demonstrating both protective and pathogenic roles. This review examines the contrasting roles of the cGAS-STING signaling pathway in intestinal inflammation and colitis-associated cancer (CAC), emphasizing its promise as a target for cancer prevention strategies. Evidence suggests that modulating this pathway could preserve epithelial integrity, limit chronic inflammation, and bolster anti-tumor immunity. Despite advancements in therapies like mesalazine and surveillance colonoscopy programs, gaps in efficacy remain, particularly for Crohn’s disease and high-risk populations. Future research should focus on integrating cGAS-STING-targeted approaches with existing modalities to provide personalized and less invasive strategies for CAC prevention. By harnessing this pathway’s therapeutic potential, a paradigm shift in managing IBD-associated CRC may be achieved, addressing the challenges of long-term disease surveillance and prevention.

## 1. Introduction

### 1.1. IBD Epidemiology, Therapy, and Colitis-Associated Cancer Risk

Inflammatory bowel disease (IBD), including Crohn’s disease (CD) and ulcerative colitis (UC), is characterized by recurrent inflammation of the gastrointestinal tract caused by an inappropriate immune response to the gut microbiome [1,2]. Globally, IBD affects approximately 90 individuals per 100,000 [3]. Over the past 50 years, its incidence has risen steadily, spreading across the globe, particularly in newly industrialized regions such as Asia, Latin America, South America, and Africa [3,4]. The therapy for IBD focuses on inducing and maintaining remission, preventing complications, and improving quality of life [5]. Induction therapy typically involves corticosteroids for active inflammation, with biologics like anti-TNF agents (e.g., infliximab, adalimumab) reserved for moderate to severe cases or patients unresponsive to conventional treatments [1,2]. Maintenance of remission relies on immunosuppressants such as thiopurines or methotrexate, as well as biologics, which are highly effective in sustaining disease control. Treatment is tailored to disease severity, location, and individual patient factors, with monitoring through biomarkers and imaging to guide adjustments [6].

Chronic inflammation is the main driver of CRC in IBD. The risk of CAC increases with the duration and extent of colonic inflammation, while limited rectal involvement (proctitis) poses no significant risk [7]. Patients with extensive ulcerative colitis face a greater risk of developing advanced neoplastic lesions than those with disease limited to the left side of the colon [8]. While advancements in endoscopic imaging, surveillance programs, and medical therapies have reduced the prevalence of CRC in IBD patients, the risk remains significant [9]. Data from 2014 indicate that the incidence of colorectal cancer among individuals with IBD was 0.91 cases per 1000 patient-years during the first decade, increasing to 4.07 in the second decade and 4.55 in the third [10]. Notably, IBD-associated CRC often demonstrates more aggressive behavior and lower rates of successful surgical resection, leading to poorer outcomes compared to sporadic CRC [11]. A Scandinavian cohort study of over 96,000 UC patients reported a 66% increased risk of CRC incidence (HR 1.66) and a 59% increased risk of CRC-related death (HR 1.59) compared to matched controls [12]. Despite being diagnosed at earlier stages, UC patients remained at higher mortality risk even after tumor stage adjustment (HR 1.54), though excess risks declined in recent years.

This highlights the need for further research into the molecular mechanisms driving IBD-related CRC (IBD-CRC) and the identification of novel therapeutic targets, with emerging evidence pointing to the cGAS-STING pathway as a key player in both human and experimental IBD models [13]. There has been a recent surge in the literature examining the cGAS-STING pathway in intestinal health and disease, reflecting its emerging importance in mucosal immunology and inflammation-driven tumorigenesis [14,15]. Recent reviews explored its role in intestinal homeostasis, its dual protective and pathogenic functions in IBD, and the influence of microbial and metabolic triggers. Others focused on its pharmacological modulation in systemic inflammatory disorders or summarized experimental therapeutics and microbiota interactions in gut pathology [16]. However, these works either lacked a dedicated focus on the inflammatory-to-neoplastic transition seen in colitis-associated colorectal cancer (CAC) or did not critically synthesize mechanistic insights from preclinical models. This review addressed that gap by examining how cGAS-STING activation contributes specifically to IBD-related colorectal carcinogenesis and discusses its potential as a chemopreventive target in high-risk patient populations. This review explored the STING pathway’s contribution to IBD pathogenesis and its role in inflammation-associated malignancies, proposing its targeting as a potential chemopreventive strategy for reducing CRC risk in IBD patients.

This narrative review was informed by a targeted literature search across PubMed, Scopus, and Web of Science up to March 2025. Search terms included combinations of “cGAS-STING”, “colitis-associated colorectal cancer”, “IBD”, “intestinal inflammation”, “epithelial barrier”, and “tumorigenesis”. We focused on original research and authoritative reviews that explored the mechanistic, preclinical, and translational aspects of cGAS-STING signaling in the context of IBD-related carcinogenesis. Particular relevance was given to studies employing murine models of colitis, which represent the primary in vivo tools for examining the immunopathology of intestinal inflammation and its malignant transformation. Given the heterogeneity of experimental designs and the evolving nature of the field, a systematic review was not deemed appropriate.

### 1.2. The cGAS-STING: Linking DNA Sensing to Immune Signaling and Function

The innate immune system in mammals employs pattern-recognition receptors (PRRs) to detect foreign genetic material by recognizing pathogen-associated molecular patterns (PAMPs) and damage-associated molecular patterns (DAMPs). These receptors initiate immune responses, including cytokine and type-I interferon (IFN) production, to fight infections and maintain cellular homeostasis [17]. Cyclic GMP-AMP synthase (cGAS), a key PRR, activates the stimulator of interferon genes’ (STING) pathway upon detecting cytosolic DNA [18]. STING, localized in the endoplasmic reticulum (ER), facilitates the production of type-I IFN once activated by bacterial ligands or cyclic GMP-AMP (cGAMP) synthesized by cGAS. This pathway plays a pivotal role in immune regulation and interacts with other DNA-sensing mechanisms to preserve cellular integrity [18].

The cGAS is a nucleotidyl transferase that senses double-stranded DNA (dsDNA) and is composed of an unstructured N-terminal domain (NTD) and a well-defined C-terminal domain (CTD), which harbors both its DNA-binding and enzymatic sites [19]. Upon binding dsDNA longer than 40 base pairs, cGAS undergoes conformational changes, catalyzing the conversion of GTP and ATP into cGAMP [20]. This molecule activates STING, driving immune responses. In the cytosol, cGAS detects DNA from various sources, including mitochondrial and microbial origins, while nuclear cGAS, bound to chromatin, can impede DNA replication and repair [21]. STING, a key ER protein, comprises four regions: a cytosolic N-terminal, a transmembrane domain, a connector, and a ligand-binding domain with a C-terminal tail (CTT) [22]. In its inactive state, STING resides in the ER membrane and associates with stromal interaction molecule 1 (STIM1) [23]. Upon cGAMP binding, STING dimerizes, undergoes structural changes, and moves to the ER-Golgi intermediate compartment (ERGIC), aided by COPII, Sar1 GTPase, and ARF proteins. At this location, STING facilitates the recruitment of TANK-binding kinase 1 (TBK1), resulting in its phosphorylation at the C-terminal tail (CTT) and promoting the assembly of the STING signalosome [23]. This activates interferon regulatory factor 3 (IRF3), which promotes type-I IFN production and the expression of interferon-stimulated genes (ISGs) via the JAK-STAT pathway [24]. Simultaneously, STING activation induces NF-κB signaling, enhancing pro-inflammatory cytokine production [25]. The above is illustrated in Figure 1.

In addition to driving transcriptional responses, STING promotes autophagy, facilitating the clearance of pathogens and damaged organelles [26]. Autophagosomes degrade STING and cGAS, creating a feedback loop to regulate their activity [27]. Post-translational modifications, such as phosphorylation, ubiquitination, and glutamylation, fine-tune cGAS-STING signaling. For instance, the ULK1-mediated phosphorylation of STING at Ser366 marks it for degradation, while ubiquitination can either enhance or suppress activity [28]. Other regulators, such as DNase II, TREX1, and AIM2, help limit cytosolic DNA to prevent overactivation of the pathway [29]. Recent studies also highlight the role of microRNAs, including miR-210 and miR-24-3p, in modulating STING expression at the translational and protein levels [30,31]. Positive regulation, through interactions and modifications such as phosphorylation at critical sites (e.g., S358, Ser366, Y245), supports robust immune responses [32].

## 2. The cGAS-STING Pathway: Bridging Innate Immunity and Chronic Diseases

The cyclic GMP-AMP synthase (cGAS) can detect and respond to both genomic and mitochondrial DNA from the host as it binds to DNA sequences longer than 45 nucleotides [15]. This ability to recognize diverse DNA sources underpins the involvement of the cGAS-STING pathway in a broad spectrum of autoinflammatory, autoimmune, and neurodegenerative diseases affecting various organs, including the brain, heart, kidneys, skin, and intestines [33]. Since nucleic acids serve as primary activators of the cGAS-STING pathway, its dysregulation is implicated in several autoimmune and autoinflammatory disorders [34]. For example, chronic activation of the pathway due to enzymatic mutations or cellular stress drives persistent type-I interferon (IFN-I) responses in systemic lupus erythematosus (SLE) [35], Sjögren syndrome [36,37,38], and scleroderma [39]. The cGAS-STING pathway is also implicated in certain rare monogenic autoinflammatory diseases. Mutations in the STING1 gene cause STING-associated vasculopathy with onset in infancy (SAVI), characterized by recurring fevers and skin lesions [40]. Similarly, Aicardi–Goutières syndrome (AGS) and COPA syndrome are linked to aberrant cGAS-STING activation, leading to systemic autoinflammatory symptoms [41,42].

The cGAS-STING pathway plays a significant role in the context of infectious diseases. DNA-containing viruses commonly activate cGAS; STING-deficient mice exhibit heightened susceptibility to such infections, highlighting its critical role in antiviral immunity. For example, the hepatitis B virus may infect human and murine hepatocytes, which often lack STING expression [43]. Remarkably, even RNA viruses like SARS-CoV-2 and HIV activate the cGAS pathway through mechanisms independent of DNA. In SARS-CoV-2 infections, mitochondrial damage and chromatin released into the cytoplasm trigger cGAS activation [44], whereas HIV activates cGAS via DNA intermediates and protein interactions [45]. In metabolic diseases, cGAS-STING activation in hepatocytes following mitochondrial DNA release promotes inflammation and cytokine production [46]. Chronic activation is associated with liver inflammation, steatosis, and metabolic dysfunction, including insulin resistance and dyslipidemia. Studies in mice demonstrate that STING inhibition reduces these metabolic disturbances, suggesting its involvement in fat and carbohydrate metabolism [46].

The cGAS-STING signaling pathway has increasingly been recognized for its involvement in cancer development [47]. Cytosolic DNA, derived from genomic instability or mitochondrial dysfunction, activates cGAS-STING signaling, which influences both tumor suppression and progression. While its activation can elicit anti-tumor immunity through type-I interferon (IFN) responses and immune cell priming, it also fosters tumor-promoting effects by facilitating metastasis, immune evasion, and chronic inflammation [47]. The balance between beneficial and detrimental outcomes depends on the duration and intensity of cGAS-STING activation, with chronic activation often leading to inflammation and tumor-supportive conditions [47]. Overall, the cGAS-STING axis is a key regulator of immune activity and inflammatory processes in various disease settings. Further research is essential to elucidate the mechanisms underlying its diverse effects, particularly in therapeutic contexts targeting immune regulation and disease amelioration.

## 3. The cGAS-STING Pathway in IBD: Integrating Innate Immunity, Inflammation, and Gut Homeostasis

The cGAS-STING signaling pathway has been identified as a fundamental pattern recognition and effector mechanism in the innate immune system [15]. It is deeply implicated in the pathogenesis, progression, and therapeutic response of IBD [15]. Evidence supporting these findings is derived from studies on IBD patients, genomic data, cell cultures, and animal models [48]. Among the available models, the dextran sulfate sodium (DSS)-induced colitis model is commonly employed. In this model, DSS administered through drinking water induces acute colitis in mice, characterized by bloody diarrhea, ulceration, tissue damage, and neutrophil infiltration, predominantly in the left colon [48]. Prolonged DSS exposure leads to chronic inflammation marked by macrophage and CD4^+^ T cell infiltration, fissuring ulcers, and increased levels of pro-inflammatory cytokines (e.g., IL-2, IL-4, IL-6), indicative of an adaptive immune response. Although not fully representative of human colitis, this model mimics critical features, such as epithelial barrier dysfunction, cytokine imbalances, and mucosal pathology [48]. Moreover, when combined with azoxymethane, DSS-induced colitis provides a robust model for studying colorectal cancer progression [49].

### 3.1. Effect of the cGAS-STING Pathway in Epithelial Cells in IBD

The cGAS-STING pathway plays a critical role in the regulation of intestinal epithelial integrity, inflammation, and immune–microbial interactions in IBD. In epithelial cells, this DNA-sensing axis can function as both a driver of barrier dysfunction and a mediator of mucosal protection, depending on the cellular context and upstream stimuli.

Loss of the DNA/RNA helicase DHX9 in intestinal epithelial cells (IECs) promotes genomic instability and the accumulation of R-loop structures—RNA–DNA hybrids that leak into the cytoplasm. This leakage activates cytosolic DNA sensors including cGAS and TLR3, leading to robust STING-mediated type-I interferon responses [50]. In mouse models, DHX9 deficiency resulted in impaired intestinal stem cell (ISC) and Paneth cell populations, epithelial barrier damage, and exacerbated colitis. Pharmacological inhibition of STING in these mice partially restored ISC and goblet cell numbers and attenuated disease severity, implicating cGAS-STING as a key mediator of inflammation-linked epithelial dysfunction [50]. In parallel, evidence from UC patients shows that STING signaling is upregulated in both IECs and lamina propria immune cells. Studies using colonic organoids showed that the combined action of IFN-β and TNF-α strengthens STING pathway activation, leading to the increased expression of PYHIN inflammasomes like IFI16, while reducing levels of protective factors such as AIM2. This cytokine milieu drives epithelial cytotoxicity, as shown by increased membrane blebbing and cell death [51].

Modulatory pathways counteract STING-driven inflammation. The IGF2/IGF2R axis was identified as a key suppressor of cGAS-STING activity. In DSS-induced colitis models, IGF2 levels were reduced, but recombinant IGF2 restored IGF2R expression, blocked cGAS-STING signaling, preserved tight junction proteins (ZO-1, occludin), and promoted epithelial regeneration [52]. These findings suggest that IGF2 exerts dual anti-inflammatory and barrier-stabilizing effects. Conversely, environmental factors can activate pathogenic STING responses. Exposure to cytotoxic epoxy triglycerides (EGTs), commonly found in repeatedly heated cooking oils, triggers cGAS-STING-NF-κB signaling in IECs, promoting pyroptosis, cytokine release, and barrier permeability—all of which contribute to disease exacerbation [53].

Beyond its canonical cytosolic role, STING also exerts nuclear functions. Zhang et al. reported that STING1 translocates to the nucleus, where it interacts with the aryl hydrocarbon receptor (AHR) through its cyclic dinucleotide-binding domain (CBD). This STING–AHR complex recruits promyelocytic leukemia (PML) nuclear bodies and enhances the transcription of IL-22 and other protective genes [54]. This nuclear interaction supports microbiota balance and epithelial recovery, whereas its absence leads to impaired AHR signaling, dysbiosis, and aggravated colitis.

However, STING signaling may also exacerbate colitis under certain conditions. Martin et al. [55] showed that activation of STING using the agonist DMXAA in wild-type mice significantly worsened DSS-induced colitis, leading to greater weight loss, epithelial injury, and inflammatory infiltration. In contrast, STING-deficient mice (Tmem173^gt) exhibited reduced disease severity, preserved epithelial architecture, and fewer activated macrophages. These results highlight the pro-inflammatory potential of excessive STING activation during acute colitis [55]. STING is also essential for microbial defense. In STING-deficient mice infected with *Citrobacter rodentium*, colitis severity increased due to impaired production of the antimicrobial peptide REG3γ. IECs lacking STING failed to restrict bacterial growth, while STING agonists restored bacterial clearance and barrier protection [56]. Similarly, STING deficiency impairs IgA production, reduces IL-1β levels, and depletes regulatory T cells (Tregs), predisposing mice to severe epithelial damage and dysbiosis across colitis and infection models [57]. STING-deficient mice also display microbial shifts favoring Proteobacteria (e.g., *Desulfovibrio*) and loss of beneficial genera like *Bifidobacterium* and *Allobaculum* [57].

Altogether, these findings illustrate that the cGAS-STING pathway is a double-edged sword in epithelial biology: while it is essential for pathogen defense, barrier integrity, and immune regulation, its overactivation can drive epithelial dysfunction and chronic inflammation. This duality makes it a compelling but complex therapeutic target in IBD and inflammation-associated carcinogenesis. The above is illustrated in Figure 2.

### 3.2. Effect of the STING Pathway in Macrophages and Dendritic Cells in IBD

STING activation in macrophages and dendritic cells (DCs) plays a pivotal role in shaping the immune microenvironment during intestinal inflammation. This axis regulates macrophage polarization, cytokine production, and microbial sensing, acting as a critical node in innate immunity and mucosal homeostasis.

Martin et al. [55] showed that STING is highly expressed in pro-inflammatory M1 macrophages and promotes their activation during DSS-induced colitis. STING agonism in wild-type mice intensified weight loss, epithelial damage, and inflammatory cytokine production (e.g., IFN-β, CXCL10, IL-12p40), whereas STING-deficient (Tmem173^gt) mice showed reduced disease severity, attenuated tissue damage, and diminished macrophage infiltration. These results emphasize the role of STING in driving macrophage polarization toward a disease-promoting phenotype in colitis. Complementary findings by Cai et al. [58] demonstrated that the myeloid-specific deletion of STING attenuates both acute and chronic colitis by inhibiting IL-12 family cytokine production in DCs. STING activation in these cells—triggered by TFAM-bound mitochondrial DNA (mtDNA) from damaged epithelial cells—promotes Th1 and Th17 differentiation via IRF3 and NF-κB signaling, perpetuating inflammation. The study underscores the significance of mitochondrial DAMPs and STING-mediated DC activation in the pathogenesis of IBD.

Shmuel et al. [59] identified that dysbiosis, particularly involving Gram-negative bacteria, promotes the K63-linked ubiquitination and stabilization of STING in myeloid cells, leading to chronic STING hyperactivation, spontaneous colitis, and fibrosis. STING accumulation was most prominent in intestinal macrophages and monocytes. Interventions targeting Gram-negative bacteria or *Helicobacter typhlonius* alleviated inflammation and reduced STING protein levels, indicating a key link between the microbiota and STING signaling [59]. These findings provide insights into the role of the microbiota–STING axis in regulating intestinal homeostasis and inflammation, offering potential therapeutic targets for treating IBD. Building on this, Ahn et al. [60] further explored this interface, showing that STING orchestrates both pro- and anti-inflammatory responses. In macrophages, it drives IL-1β, IL-18, and IL-22 production—cytokines essential for antimicrobial defense and epithelial repair—but also induces IL-10, which limits excessive inflammation. STING deficiency led to unbalanced cytokine responses, enhanced pathology, and impaired resolution, demonstrating its dual regulatory role [60].These findings underscore the importance of macrophage-driven STING activity in orchestrating intestinal immune balance, providing critical insights into its therapeutic potential for managing inflammatory bowel diseases.

Extracellular vesicles (EVs) are another potent activator of STING. Zhao et al. [61] reported that EVs released from damaged epithelial cells in CD carry nuclear and mitochondrial DNA, activating STING in macrophages and driving the secretion of IL-6, TNF-α, and IFN-β. STING-deficient macrophages or blockade of EV release significantly reduced disease severity, implicating EV-carried dsDNA as a novel driver of STING-mediated inflammation in CD. Extending this, Nie et al. [62] found that microbial EVs (mEVs) containing bacterial DNA accumulate in IBD due to a deficiency of CRIg^+^ macrophages responsible for their clearance. In CRIg knockout mice, mEV buildup triggered epithelial dysfunction and inflammation via STING activation. Reinstating CRIg^+^ macrophage activity or pharmacologically inhibiting STING reduced disease severity, underscoring the critical role of EV–STING signaling in maintaining host–microbe equilibrium. Collectively, these studies reveal the central role of STING in myeloid cells in coordinating immune responses, linking epithelial damage and dysbiosis to chronic inflammation. Depending on the stimulus and timing, STING can drive tissue-destructive immunity or modulate protective feedback mechanisms. Therapeutic strategies aimed at selectively modulating STING in macrophages and DCs could offer promising avenues to attenuate IBD without compromising mucosal defense.

### 3.3. Effect of the STING Pathway in B and T Lymphocytes in IBD

STING signaling plays a crucial role in modulating the activity and differentiation of lymphocytes, particularly CD4^+^ T cells, during inflammatory conditions such as IBD. In T lymphocytes, STING expression is particularly elevated in Th1 cells, where it serves to regulate cytokine production and influence the inflammatory milieu [63].

Yang et al. [63] demonstrated that STING reduces Th1 cell pathogenicity by promoting an IFNγ^+^IL-10^+^ phenotype through STAT3-dependent Blimp1 expression and mitochondrial metabolism. Mice lacking STING exhibited more severe colitis, characterized by reduced IL-10 production and elevated levels of TNF-α and IL-6. Treatment with STING agonists (DMXAA, 2,3-cGAMP) improved disease outcomes by restoring IL-10 in Th1 cells. Similar effects were observed in human CD4^+^ T cells, independent of microbiota composition. Shmuel et al. [59] demonstrated that excessive STING activation promotes the expansion of pro-inflammatory TCRβ^+^ Th1 cells and reduces regulatory T cells (Tregs), thereby worsening colitis. TCRβ^+^ T cells were essential for disease development in their model, while TCRδ^+^ T cells were dispensable. Colonic CD4^+^ T cells displayed resistance to apoptosis, maintaining an inflammatory niche that supported dysbiosis—especially overgrowth of Gram-negative bacteria like *Helicobacter typhlonius* [59]. In summary, STING activation indirectly promotes an inflammatory T cell response in the colon by expanding Th1 cells, reducing Tregs, and fostering an environment conducive to dysbiosis. These findings highlight the central involvement of T cells in the advancement of STING-mediated intestinal inflammation.

As mentioned above, in a study by Cai et al., damaged intestinal epithelial cells released TFAM-associated mtDNA, which activated the STING pathway in myeloid-derived DCs [58]. This activation promoted the production of IL-12 family cytokines, which in turn primed and enhanced the differentiation of pro-inflammatory Th1 and Th17 cells. These T cells played a role in the worsening of colitis by perpetuating the inflammatory process. The findings underscore the role of STING-mediated signaling in linking innate and adaptive immunity, particularly through its impact on T cell differentiation and inflammatory cytokine production, offering insights into potential therapeutic strategies for ulcerative colitis and related conditions [58]. Finally, Damasceno et al. investigated the impact of the STING agonist 2′3′-RR CDA on human TH17 cells, revealing its ability to reduce IL-17A production while promoting the conversion of TH17 cells into Tregs [64]. These results pointed to STING activation as a promising therapeutic approach for autoimmune conditions characterized by TH17-driven inflammation. The activation of STING was shown to reduce IL-17A production by TH17 cells while preserving expression of their lineage-defining factor, Rorγt. Additionally, it promoted a shift in the TH17/Treg balance towards a more anti-inflammatory state, suggesting an important regulatory role for STING in immune responses [64]. The results underline the potential of STING agonists like 2′3′-RR CDA to treat chronic inflammatory conditions by reprogramming TH17 cells. The study emphasized STING’s adaptability in regulating immune responses and its therapeutic promise for TH17-driven diseases.

In summary, STING governs lymphocyte responses in IBD by repressing inflammatory programs and promoting regulatory phenotypes. It shapes Th1/Th17 polarization, enhances IL-10 production, and contributes to adaptive immunity–microbiota feedback loops, offering a promising immunomodulatory target in IBD.

### 3.4. Role of Sting in Neuroinflammation in IBD

The enteric nervous system (ENS), composed of neurons and glial cells, plays a pivotal role in regulating gastrointestinal functions such as motility, secretion, and immune responses [65,66]. STING signaling within the ENS has emerged as a key factor in modulating neuroinflammation [67] and maintaining gut homeostasis in the context of IBD. Enteric neurons and glial cells both express STING, although their functional roles differ.

Dharsika et al. explored the role of STING in the ENS and its involvement in gastrointestinal inflammation [68]. Using genetic mouse models, transcriptional analyses, and immunohistochemistry, the researchers demonstrated that both enteric neurons and glial cells express STING. However, IFNβ production is primarily localized to neurons, while glial STING contributes minimally to IFNβ signaling, suggesting a distinct functional role. They showed that STING activation induced IFNβ production in neurons of the myenteric and submucosal plexuses. In contrast, glial STING activation is primarily associated with autophagy, characterized by increased LC3 expression independent of IFNβ. This highlights an alternative, non-canonical role for glial STING in cellular processes [68]. In a model of acute DSS-induced colitis, the deletion of glial STING did not significantly affect weight loss, colonic shortening, histological damage, or neuronal composition, indicating that glial STING played a limited role in acute inflammatory responses. These findings suggest that glial STING may function in more specialized or chronic conditions, potentially regulating neuroimmune interactions or modulating inflammation through autophagy mechanisms [68]. Overall, the study highlights the complex, cell-specific roles of STING in the ENS, with neurons driving IFNβ-mediated immune responses and glial cells engaging in alternative signaling pathways like autophagy. Further investigation is needed to understand the contexts in which glial STING contributes to gastrointestinal health and disease. A recent study highlighted the role of the neuropeptide Substance P (SP) in modulating STING activity during colitis [69]. SP, which is released by enteric neurons, was shown to exert protective effects on the intestinal barrier. It enhanced mucus production, promoted the expression of tight junction proteins (e.g., occludin, ZO-1), and reduced epithelial permeability, all of which are critical for preventing intestinal damage during inflammation. Moreover, SP suppressed the cGAS-STING pathway by downregulating key signaling proteins, including cGAS, STING, p-TBK1, and p-IRF3. This suppression reduced the production of pro-inflammatory cytokines such as IL-6, TNF-α, and IL-1β, thereby mitigating the inflammatory response. SP also promoted the polarization of macrophages toward an M2 phenotype. In addition to its effects on barrier integrity and macrophage polarization, SP was found to protect intestinal cells from ferroptosis, a type of programmed cell death associated with oxidative stress. Even in the presence of STING agonists, SP significantly reduced ferroptosis markers and preserved intestinal architecture, further emphasizing its protective role [69].

STING signaling in the enteric nervous system supports gut homeostasis and modulates neuroinflammation in IBD. While neurons primarily drive immune responses through IFNβ production, glial STING appears to function in cellular processes like autophagy, with potential roles in chronic or specialized conditions. Additionally, Substance P regulates STING activity, supporting intestinal barrier function and reducing inflammation.

## 4. The cGAS-STING Pathway in Colitis-Associated Carcinogenesis: From Pathogenesis to Treatment

Clinical evidence regarding the involvement of the cGAS-STING pathway in the development of colitis-associated colorectal cancer (CAC) remains limited. However, Wang et al. conducted a comprehensive study exploring the role of this pathway in the transition from UC to CAC, providing valuable insights into its dual function in inflammation and tumorigenesis [70]. While the pathway was highly activated in UC, driving immune and inflammatory responses, its activity decreased during the transition to CAC. This shift in pathway activity suggested a dual role, where it initially promotes inflammation but later supports anti-tumor immunity. A key gene, IFI16, identified as a hub gene in the cGAS-STING pathway, plays a crucial role in this dynamic. IFI16 expression was elevated during the inflammatory phase of UC, correlating with an increased immune burden and heightened inflammation. However, IFI16 expression decreases as the disease progresses to CAC. This reduction in IFI16 and overall cGAS-STING pathway activity aligned with decreased immune infiltration and an impaired anti-tumor immune response, potentially contributing to the development of CAC. Interestingly, in sporadic colorectal cancer and CAC, high IFI16 expression was linked to better overall survival and favorable outcomes in patients receiving immunotherapy. This association highlights IFI16’s role in maintaining immune surveillance and fostering an anti-tumor environment. The study underscores that tumors with robust immune infiltration, often referred to as “hot tumors”, are more responsive to immunotherapy, with IFI16 serving as a marker of such immune activity [70].

### 4.1. Epithelial Protection and Immune Surveillance: cGAS-STING’s Role in CAC Pathogenesis

Despite limited clinical evidence, there has been a surge in preclinical research exploring the role of the cGAS-STING pathway in the development and progression of CAC, highlighting its profound effects on cellular and molecular pathways involved in inflammation, immune surveillance, and tumorigenesis. Epithelial integrity plays a vital role in CAC, serving as a critical barrier that regulates immune responses and prevents chronic inflammation-driven tumorigenesis [71,72]. Hu et al. examined the role of the cGAS enzyme in preserving intestinal barrier function and preventing CAC. They demonstrated that the absence of cGAS exacerbated inflammation, compromised the intestinal epithelial barrier, and accelerated the development of CAC in mouse models [71]. The cGAS has been found to influence intestinal stem cell function and maintain epithelial barrier integrity independently of STING signaling. Mice lacking cGAS exhibited severe colitis and increased tumor burden when exposed to inflammation-induced carcinogenesis. The loss of cGAS led to intestinal stem cell depletion, increased epithelial cell proliferation, and compromised barrier function, evidenced by enhanced permeability and the reduced expression of mucins. This barrier dysfunction fueled chronic inflammation, marked by the elevated production of cytokines such as IL-6 and activation of STAT3, a signaling pathway strongly associated with colorectal cancer development [71]. Additionally, cGAS deficiency resulted in an inflammatory tumor microenvironment characterized by the accumulation of myeloid-derived suppressor cells (MDSCs) and Th17 cells, which contributed to immune suppression and tumor progression. These mice also exhibited T cell exhaustion within tumors, further impairing anti-tumor immune responses [71]. Importantly, the protective role of cGAS was independent of the STING and type-I interferon pathways, as mice lacking STING or type-I interferon receptor did not show similar susceptibility to CAC. Moreover, therapeutic intervention using cGAMP, the product of cGAS activity, partially mitigated tumor development, highlighting its potential for treating CAC [71]. The study underscored the critical role of cGAS in maintaining intestinal homeostasis by preserving the epithelial barrier and modulating immune responses [71]. Analogously, Zhu et al. demonstrated that STING protects against CAC by modulating inflammation and limiting tumor-promoting signaling [72]. In a mouse model using AOM and DSS, STING-deficient mice exhibited increased tumor burden, severe intestinal damage, and heightened inflammation. These mice showed elevated levels of pro-inflammatory cytokines IL-6 and KC, along with enhanced NF-κB and STAT3 signaling, which promote tumorigenesis through inflammation and epithelial proliferation. Additionally, STING deficiency reduced caspase-1 activation and IL-18 release, disrupting protective immune responses at the intestinal barrier [72]. Despite no significant changes in gut microbiota composition, STING-deficient mice displayed amplified crypt hyperplasia and dysplasia, further highlighting the pathway’s role in suppressing tumor initiation. The findings suggest that STING regulates key molecular pathways to maintain intestinal homeostasis and prevent inflammation-driven CRC, with potential therapeutic implication. Taking a step further, Ahn et al. demonstrated that STING promotes the production of IL-18, which plays a pivotal role in suppressing the expression of interleukin-22 binding protein (IL-22BP) [73]. IL-22BP is a regulatory protein that inhibits the activity of interleukin-22 (IL-22), a cytokine responsible for stimulating the proliferation of intestinal epithelial cells and facilitating tissue repair following damage [74]. In the absence of STING, IL-18 production was reduced, leading to elevated levels of IL-22BP. This suppression of IL-22 activity delayed tissue regeneration, prolonged inflammation, and increased tumor susceptibility in the colon. The findings also indicated that, while IL-22 promotes tissue repair, its unchecked activity, due to inadequate regulation by IL-22BP, can exacerbate tumorigenesis. STING’s regulation of the IL-18/IL-22BP axis is therefore critical for maintaining a balance between inflammation resolution and preventing tumor development. In STING-deficient mice, this regulatory balance was disrupted, resulting in excessive intestinal inflammation and an enhanced risk of colorectal cancer progression [73]. These results underscore the central role of interleukins, particularly IL-18 and IL-22, in the STING-mediated protective mechanisms against inflammation-driven cancers like CAC. The above is illustrated in Figure 3.

### 4.2. The cGAS-STING Signaling in CAC: A Therapeutic Focus on Inflammation Control

Emerging evidence highlights the critical role of the cGAS-STING pathway in orchestrating the transition from chronic inflammation to CAC [60,75,76]. Ghonim et al. provides compelling evidence regarding CAC using inflammation-driven models, specifically the azoxymethane (AOM)/dextran sulfate sodium (DSS) [75]. In the AOM/DSS model, the partial inhibition of PARP-1 through genetic heterozygosity, which reduces PARP-1 levels by approximately 50%, significantly decreased tumor burden. Mice with PARP-1 heterozygosity exhibited smaller tumors primarily confined to aberrant crypt foci (ACFs) and demonstrated reduced proliferation markers such as PCNA compared to wild-type mice. Additionally, these mice displayed better mucosal integrity and a reduction in colon inflammation. The complete inhibition of PARP-1, however, was less effective. PARP-1 knockout mice exhibited some protection against colitis but failed to significantly reduce tumor burden, with an aggravation of inflammatory markers. The pharmacological inhibition of PARP-1 using olaparib mirrored these findings. Low to moderate doses of olaparib (5 mg/kg) effectively reduced tumor burden and colitis, achieving results comparable to PARP-1 heterozygosity. However, high doses (25 mg/kg) showed variable outcomes and were less effective in preventing tumor progression and colitis. PARP-1 inhibition also reduced levels of key pro-inflammatory cytokines, such as TNF-α and MCP-1, which are critical in the pathogenesis of colitis-associated cancer. This led to a significant reduction in systemic inflammation [75]. Mechanistically, the beneficial effects of partial PARP-1 inhibition were attributed to the altered activity of myeloid-derived suppressor cells (MDSCs), dampened pro-inflammatory pathways—including TNF-α and IL-6—and enhanced preservation of the mucosal barrier. Excessive PARP-1 inhibition, on the other hand, led to the dysregulation of immune responses and amplified inflammation, contributing to tumor progression. In conclusion, the findings indicate that the partial inhibition of PARP-1, whether through genetic or pharmacological means, provides robust protection against colitis-associated colorectal cancer [75]. This protection is mediated by reducing inflammation, promoting mucosal healing, and suppressing tumor development. Ahn et al. investigated also the molecular mechanisms through which the STING pathway influences CAC [60].

As previously noted, STING plays a central role in innate immunity, becoming activated by cyclic dinucleotides (CDNs), like c-di-AMP and c-di-GMP from bacterial sources, or by the endogenous molecule cGAMP, which is synthesized by cGAS in response to microbial or host-derived DNA. In models of CAC induced by azoxymethane (AOM) and dextran sodium sulfate (DSS), STING-deficient (SKO) mice developed a higher number of colonic polyps compared to wild-type mice, implicating STING signaling in suppressing tumor-promoting inflammation. Antibiotic treatment significantly reduced polyp formation in SKO mice, indicating that commensal bacteria and their metabolites play a pivotal role in modulating STING activity. Further analysis revealed that STING-deficient mice exhibited dysbiosis, with altered profiles of bacterial species such as Turicibacter and Odoribacter, which may exacerbate inflammation and tumorigenesis [60]. STING signaling was shown to regulate the production of both pro-inflammatory and anti-inflammatory cytokines, establishing a balance that influences tumor progression. The absence of STING led to decreased levels of the anti-inflammatory cytokine IL-10, which is essential for suppressing inflammation-driven carcinogenesis. Simultaneously, STING deficiency resulted in the dysregulated production of pro-inflammatory cytokines, including IL-1β and IL-18, which contribute to tumor promotion by fostering an inflammatory microenvironment. These cytokines are known to signal via MyD88-dependent pathways, emphasizing the role of STING in controlling downstream inflammatory responses [60]. They also highlighted that the STING-mediated recognition of microbial DNA or CDNs plays a more prominent role in carcinogenesis than cGAS-dependent activation by self-DNA [60]. This distinction suggests that bacterial products are primary drivers of STING activation in this context. Overall, these findings establish STING as a central regulator of the inflammatory cascade that bridges chronic inflammation and colitis-associated cancer, identifying it as a potential therapeutic target for mitigating inflammation-driven tumorigenesis. Yang et al. explored the protective role of palbociclib, a cyclin-dependent kinase (CDK4/6) inhibitor and STING pathway antagonist, in CAC [76]. Using a CAC mouse model induced by AOM and DSS, they demonstrated that palbociclib alleviates colon inflammation and reduces tumor progression. Palbociclib treatment effectively mitigated epithelial damage, atypical hyperplasia, and inflammatory cell infiltration in the colon tissue of AOM/DSS-treated mice. It also significantly reduced the expression of pro-inflammatory cytokines such as *Ifnb1*, *Il6*, and *Il1b*, downstream effectors of the STING pathway, as evidenced by qRT-PCR analysis. Mechanistically, palbociclib inhibited STING activation by targeting key residues involved in dimerization and downstream signaling, thereby suppressing inflammation and tumorigenesis. These findings suggest that palbociclib exerts its protective effects on CAC by modulating the STING pathway, offering potential therapeutic value for inflammation-associated CRC [76]. Further clinical and mechanistic studies are needed to validate its efficacy and uncover additional biological markers for treatment outcomes. The above is summarized in Table 1.

These findings provide valuable insights into the molecular mechanisms underlying CAC and suggest that targeting the cGAS-STING pathway could hold therapeutic promise for preventing or treating inflammation-driven colorectal cancer. Further research is necessary to explore the precise mechanisms driving these changes and to validate the pathway’s clinical relevance in CAC management.

### 4.3. Pyroptosis and Immune Regulation: Unlocking STING’s Role in CAC Immunotherapy

According to limited evidence, targeting the cGAS-STING pathway shows potential as a novel immunotherapeutic approach in CAC, balancing inflammation modulation and anti-tumor immunity. Cai et al. extensively analyzed the role of inflammation in colitis progression above [58]. Here, we examined their findings on the effects of STING activation in CAC. They demonstrated the sophisticated role of STING activation at different disease stages. During the early inflammatory phases, STING activation exacerbated inflammation, accelerating the progression from colitis to CAC. Myeloid-specific STING deletion in the pre-inflammatory phase significantly reduces tumor formation, indicating its potential as a therapeutic target to prevent inflammation-driven tumorigenesis. However, once tumors are established, the role of STING shifts. Deleting STING at this stage suppressed cytotoxic T lymphocyte activity and reduced the infiltration and function of CD8^+^ T cells, granzyme B, and perforin. This shift resulted in an immunosuppressive tumor microenvironment, facilitating tumor growth and progression. These findings underscore the dual role of STING in the colitis-to-CAC progression. While STING activation drives inflammation and tumor initiation, it is also necessary to maintain anti-tumor immunity in established tumors. Therefore, the timing of STING-targeted therapies is critical. Targeting upstream processes, such as the release or recognition of TFAM-mtDNA by STING, might present a novel strategy to break the inflammation–carcinogenesis cycle without impairing immune responses in established tumors [58]. Thus, Gong et al. explored the molecular mechanisms through which STING mitigates CAC by regulating tumor cell pyroptosis [77]. STING functions as a cytosolic sensor of CDNs and activates downstream signaling pathways, including TANK-binding kinase 1 (TBK1) and interferon regulatory factor 3 (IRF3), to regulate type-I interferons (IFNs) and inflammatory cytokines [78]. Vasudevan et al. demonstrated that STING interacted with spleen tyrosine kinase (Syk), a non-receptor tyrosine kinase, to induce pyroptosis—a pro-inflammatory form of programmed cell death mediated by the cleavage of gasdermin D (GSDMD) [79]. Pyroptosis disrupts tumor cell integrity and releases inflammatory mediators, creating an anti-tumor microenvironment [79]. STING-deficient mice in a CAC model induced by AOM and DSS exhibited decreased GSDMD cleavage, leading to reduced tumor cell pyroptosis and enhanced tumor progression. The activation of STING using the agonist DMXAA enhanced pyroptosis by inducing the phosphorylation of Syk, which then facilitated GSDMD cleavage. They confirmed a physical interaction between STING and Syk in colorectal cancer tissues, which was crucial for Syk phosphorylation and the activation of downstream pathways. Further, the knockdown of Syk using siRNA in human colorectal cancer cells (HT-29) diminished STING-induced pyroptosis, highlighting the role of Syk as a mediator of STING’s anti-tumor effects. STING also regulated the TBK1-IRF3 pathway downstream, promoting the phosphorylation of TBK1 and IRF3 to enhance type-I IFN production. This pathway and Syk-mediated pyroptosis work synergistically to suppress tumorigenesis. Syk silencing reduced TBK1 and IRF3 phosphorylation, underscoring its bridging role in STING-mediated IFN signaling. Additionally, STING was shown to modulate the tumor microenvironment. Its deficiency increased tumor cell proliferation, as evidenced by elevated Ki67 expression, and invasiveness, with increased β-catenin and reduced E-cadherin expression. STING deficiency also altered inflammatory cytokine levels, reducing pyroptosis-related cytokines such as IL-1β and IL-18 while increasing tumor-promoting cytokines like IL-6 and TNF-α. STING activation with DMXAA suppressed tumor growth, reduced tumor numbers, and enhanced immune cell infiltration, underscoring its role in both modulating inflammation and promoting anti-tumor immunity [77]. The study highlights the protective role of STING in CAC through its ability to induce tumor cell pyroptosis and orchestrate an immune response that limits tumor proliferation and invasiveness. These findings indicate that targeting the STING-Syk axis could be a promising therapeutic approach for CAC [77]. STING agonists like DMXAA hold potential for enhancing anti-tumor immunity and pyroptosis in CAC patients, although further research is needed to investigate its potential in combination with other immunotherapies. The above is briefly summarized in Table 2.

### 4.4. Therapeutic Strategies Targeting the cGAS-STING Pathway

Modulating the cGAS-STING pathway has emerged as a promising therapeutic approach for both inflammation-driven carcinogenesis and immune modulation in CAC. Several classes of therapeutic agents are under investigation, including synthetic STING agonists, naturally derived modulators, and small-molecule inhibitors targeting various stages of the signaling cascade.

STING agonists, such as DMXAA and 2′3′-cGAMP analogs [77,80], have shown the capacity to enhance anti-tumor immunity in murine models of CAC by promoting type-I interferon production, CD8^+^ T cell recruitment, and pyroptotic cell death within tumors [77]. These effects have led to the development of human-specific agonists, including RR-CDA (MIW815) [81], SNX281, and diABZI, several of which are currently undergoing clinical evaluation [82]. On the other hand, the pro-inflammatory role of STING signaling in the early stages of colitis and CAC initiation has prompted interest in STING antagonists. Compounds such as H-151 and C-176 interfere with STING activation by preventing its palmitoylation and translocation to signaling complexes, thereby suppressing the downstream release of inflammatory mediators [83]. These antagonists have demonstrated protective effects in experimental colitis, reducing epithelial injury and dampening cytokine production. Natural compounds represent an additional area of therapeutic interest. Molecules such as andrographolide [84,85] and nitro-oleic acids have been shown to modulate STING activation indirectly by interfering with upstream triggers like oxidative stress or DNA leakage, preserving epithelial barrier integrity and ameliorating inflammation in colitis models [16]. Unlike direct inhibitors, these agents offer a broader anti-inflammatory profile with lower toxicity, making them candidates for chronic use in inflammatory bowel disease.

A major limitation in translating STING modulators into clinical practice lies in the challenges of drug delivery [14]. Many STING agonists are cyclic dinucleotides (CDNs), which are hydrophilic, negatively charged, and poorly permeable across cell membranes. As a result, systemic administration often results in suboptimal intracellular concentrations and unintended immune activation. Intratumoral injection has been used to achieve high local concentrations and limit systemic cytokine release, but this approach is not feasible in all clinical scenarios, particularly in the setting of diffuse inflammation or multifocal disease [83]. To address these issues, several formulation strategies are under development. Nanoparticle-based systems, including liposomes and polymeric vesicles, have been employed to enhance the cellular uptake and tissue specificity of CDNs [86]. Other approaches involve conjugating agonists to cell-penetrating peptides or encapsulating them in pH-sensitive carriers designed for colonic release [87]. Engineered bacterial vectors expressing STING ligands have also shown potential in selectively activating immune responses within the tumor microenvironment [88]. Despite these advances, challenges remain in achieving precise targeting, avoiding degradation by enzymes such as ENPP1, and maintaining a balance between sufficient immune activation and the risk of systemic inflammation.

Combination therapies are also under active investigation. STING agonists may synergize with immune checkpoint inhibitors, IDO blockade, or anti-angiogenic agents to amplify immune responses and overcome tumor-mediated immune evasion [81,82]. Finally, an emerging strategy involves targeting upstream events, such as the release of mitochondrial DNA from epithelial cells, which can initiate chronic STING activation and contribute to inflammation-induced carcinogenesis [63]. By modulating this axis, it may be possible to preserve the protective effects of STING in later stages of cancer while preventing its harmful activation during early inflammation.

In summary, while cGAS-STING modulation offers a promising route for immunotherapy in CAC, effective application will depend on refining delivery strategies, selecting appropriate therapeutic windows, and tailoring approaches to specific disease stages. While multiple STING-targeted agents—such as MIW815 (RR-CDA), SNX281, and diABZI—are currently being evaluated in early-phase oncology trials, it is important to clarify that no registered interventional studies have yet investigated these modulators in IBD or CAC. This absence highlights a current translational gap between promising mechanistic insights and clinical application in IBD.

## 5. Discussion

The cGAS-STING pathway presents significant therapeutic and prophylactic potential in managing CAC. Prophylactically, modulating this pathway could limit epithelial damage [71,75] and suppress chronic inflammation [60,75], which are critical in preventing the transition from colitis to CAC. For example, cGAS plays a vital role in maintaining epithelial integrity, preserving intestinal stem cells, and preventing barrier dysfunction [71]. Therapeutic interventions using cGAMP, the product of cGAS, have demonstrated efficacy in reducing inflammation and tumor burden in preclinical models. Additionally, STING’s ability to regulate interleukins, such as IL-18, highlights its role in promoting tissue repair and maintaining immune balance [60,72]. In established CAC, the pathway’s immunomodulatory capabilities could be harnessed to boost anti-tumor immunity [77]. STING activation has been shown to induce pyroptosis, a form of programmed cell death, creating an anti-tumor microenvironment by enhancing immune cell infiltration and reducing tumor cell proliferation. Agonists like DMXAA have demonstrated potential in increasing type-I interferon production and enhancing cytotoxic T cell responses [77]. However, the timing of intervention is critical, as early activation may prevent tumor initiation, while later activation could amplify immune responses to combat established tumors. Despite these promising findings, the cGAS-STING pathway’s pro-inflammatory effects pose a risk of exacerbating IBD, particularly when its activation is not tightly regulated. Excessive or poorly timed activation could worsen inflammation, contributing to intestinal damage and increasing cancer risk. Thus, strategies targeting this pathway must carefully balance its anti-inflammatory and pro-immunity roles, with further research needed to refine therapeutic approaches and validate their clinical relevance.

Patients with inflammatory bowel disease affecting the colon face a heightened risk of developing CAC [89]. This risk becomes more significant after 8–10 years of disease duration, especially in cases of extensive colitis or when high-grade dysplasia is identified during biopsies [90]. Although recent advancements in treatment and surveillance methods have led to a decrease in CRC incidence among IBD patients over the past two decades, their risk remains approximately twice that of the general population [91,92]. CRC related to IBD accounts for around 2% of overall CRC-related deaths annually and is associated with poorer survival outcomes, particularly among individuals under 50 years of age [93]. Several factors contribute to this elevated risk. The extent and duration of colonic inflammation are major determinants, with extensive colitis posing a greater risk compared to limited disease, such as proctitis [89]. Persistent histological inflammation, even when not visibly apparent during endoscopy, is closely linked to CRC development. This underscores the impact of sustained inflammation in promoting the development of cancer [94]. Additionally, primary sclerosing cholangitis (PSC), especially in UC patients, significantly increases the risk of CRC, with a predilection for right-sided colon cancers in this subgroup [95]. Male sex, early onset of UC, and a family history of CRC, particularly in first-degree relatives diagnosed at a younger age, are other important risk factors [89]. Low-grade dysplasia detected during surveillance is a critical warning sign, as it significantly raises the likelihood of progression to advanced lesions or cancer [94]. Pseudopolyps, indicating past severe inflammation, were previously associated with a higher neoplasia risk, although recent evidence questioned this correlation [96].

To address this risk, comprehensive surveillance programs were established [89]. An initial screening colonoscopy is recommended eight years after the onset of IBD symptoms, with earlier and more frequent surveillance suggested for individuals with PSC. Subsequent surveillance intervals depend on individual risk factors: annual colonoscopy for high-risk patients, every two to three years for intermediate-risk patients, and every five years for low-risk individuals. Surveillance is most effective when conducted during periods of disease remission, as active inflammation can obscure dysplastic changes. Advances in endoscopic techniques, such as high-definition imaging and chromoendoscopy, have improved the detection of dysplasia [94]. Despite the declining incidence of CRC in IBD due to improved treatments and surveillance, personalized assessment of risk factors—including disease extent, inflammatory burden, PSC, and family history—remains essential for the effective prevention and early detection of CRC in this population.

Given the challenges associated with exhaustive surveillance programs, particularly in high-risk populations requiring frequent colonoscopies, targeting the cGAS-STING pathway could provide an adjunctive approach to reduce the burden of CRC risk. Currently, aminosalicylates, particularly mesalazine, show a protective effect against CAC in UC [97], with higher doses offering greater benefits. However, evidence for their efficacy in CD is limited [98]. Despite some conflicting data, mesalazine is recommended for UC patients to lower colorectal cancer risk [99]. Thiopurines may reduce CAC risk in UC but show inconsistent results across studies [100]. Long-term use increases risks for non-melanoma skin cancer and lymphoproliferative disorders, limiting their role as standalone preventive agents [94]. Biological agents, including anti-TNF drugs, have not shown significant chemopreventive effects against CAC [94]. While they do not appear to increase CAC risk, their potential indirect benefits require further study. Incorporating novel strategies, such as targeting the cGAS-STING pathway, alongside established therapies like mesalazine, could pave the way for more effective and less invasive approaches to CAC prevention in high-risk IBD populations, complementing existing surveillance programs.

## 6. Conclusions and Future Perspectives

CAC remains a serious long-term complication of IBD, particularly in patients with extensive colonic involvement, prolonged disease duration, and additional risk factors such as primary sclerosing cholangitis. Despite advancements in surveillance and medical therapy, these patients continue to face a significantly elevated cancer risk compared to the general population. Current preventive strategies—primarily based on mesalazine, thiopurines, and intensive colonoscopic surveillance—are effective but limited and are associated with patient burden, incomplete protection, and potential long-term risks. In this context, the cGAS-STING pathway has emerged as a compelling therapeutic target with dual relevance: in the early stages of disease, it modulates epithelial barrier integrity, stem cell homeostasis, and innate immunity; in later stages, it contributes to anti-tumor immunity via pyroptosis, cytokine production, and enhancement of cytotoxic T cell infiltration. Experimental studies showed that activation of this pathway via endogenous ligands or pharmacologic agonists such as cGAMP and DMXAA can reduce tumor burden, suppress inflammation, and restore epithelial integrity. However, these effects are highly context dependent. Premature or excessive activation may worsen inflammation, while suppression could compromise immune surveillance in established tumors.

Clinical translation of cGAS-STING modulation requires a nuanced understanding of disease stage, cell-type specificity, and timing. Targeted delivery systems, including nanoparticles, mucosal vectors, and orally bioavailable compounds, may help limit systemic toxicity and off-target inflammation. Furthermore, species specificity of existing STING agonists underscores the need for human-compatible molecules that preserve efficacy while minimizing cytokine toxicity. As personalized medicine advances, incorporating specific cGAS-STING-related biomarkers—such as IFI16, MB21D1 (cGAS), TMEM173 (STING), and TBK1—may enhance treatment stratification in ulcerative colitis. Notably, IFI16 showed strong diagnostic value and may help identify patients less likely to respond to anti-TNF therapy [70]. Ultimately, integrating cGAS-STING-targeted interventions into current clinical practice has the potential to reduce CRC risk in IBD patients, particularly in those at high risk who currently require lifelong surveillance. These agents could serve as adjuncts to existing therapies, enhancing mucosal healing and dampening immune-mediated tumor suppression. Future clinical trials should focus on defining the therapeutic window, optimizing delivery, and evaluating long-term safety. If successful, such approaches may transform CAC prevention from a reactive surveillance paradigm to a proactive, mechanism-driven strategy.

## Figures and Tables

**Figure 1 ijms-26-04979-f001:**
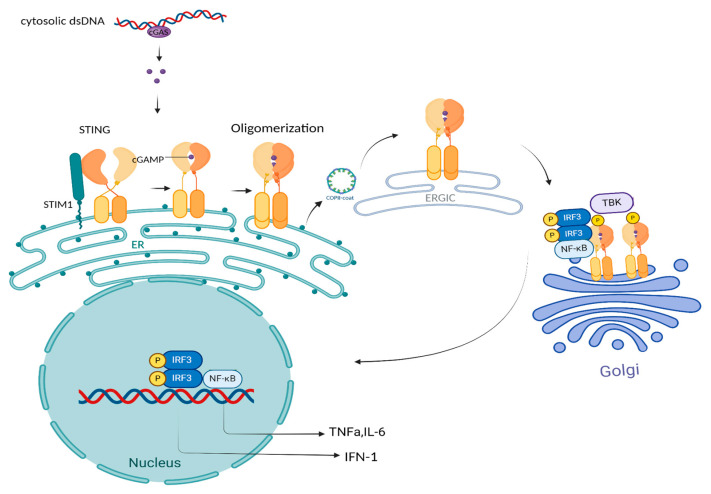
Illustrates the cGAS-STING pathway, a critical link between DNA sensing and immune signaling. When cGAS recognizes cytosolic double-stranded DNA, it triggers the production of cGAMP, a second messenger that subsequently activates the endoplasmic reticulum-resident protein STING. This activation triggers a cascade involving TBK1 and IRF3, leading to type-I IFN and pro-inflammatory cytokine production to maintain immune homeostasis. Created in BioRender. Papadakos, S. (2025) https://BioRender.com/n3mootb (accessed on 6 April 2025).

**Figure 2 ijms-26-04979-f002:**
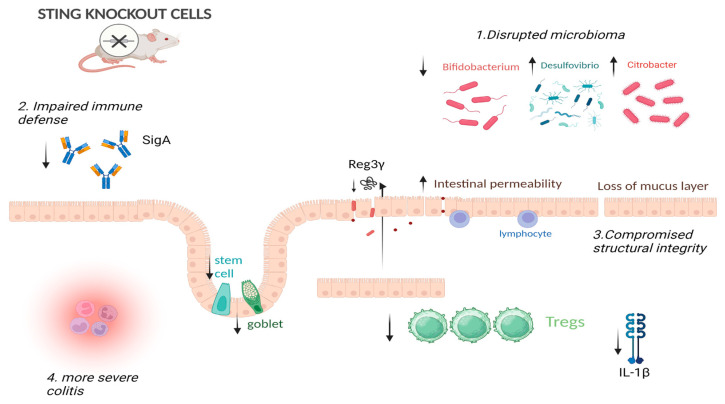
This illustrates the multifaceted impact of STING signaling on intestinal homeostasis, including permeability, microbiota balance, mucus layer integrity, stem cell function, and goblet cell populations. STING depletion disrupts epithelial tight junctions, increasing permeability and promoting dysbiosis. It disrupts the regeneration of intestinal stem cells and impairs goblet cell function, leading to compromised integrity of the mucus barrier. Therapeutic targeting of STING signaling has shown potential to restore these functions and mitigate intestinal inflammation. Created in BioRender. Papadakos, S. (2025) https://BioRender.com/ekw7eio (accessed on 18 May 2025).

**Figure 3 ijms-26-04979-f003:**
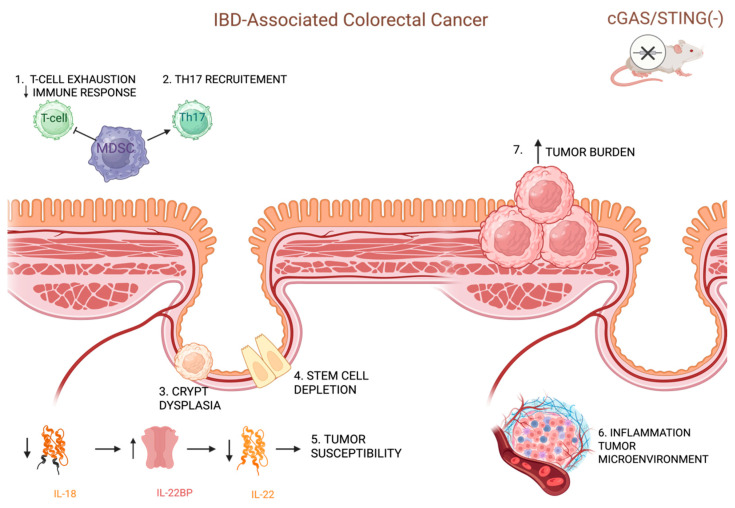
Tumor-suppressive roles of cGAS and STING signaling in inflammation-driven colorectal cancer. Loss of cGAS impairs epithelial barrier integrity, promotes chronic inflammation, and increases tumor burden via STAT3 activation and immune suppression. STING deficiency enhances pro-inflammatory cytokines, reduces IL-18 and IL-22 regulation, and disrupts tissue repair, collectively driving CAC progression. Created in BioRender. Papadakos, S. (2025) https://BioRender.com/uty1xz9 (accessed on 14 May 2025).

**Table 1 ijms-26-04979-t001:** Summarizes the key findings from three studies investigating the molecular mechanisms and therapeutic implications of the cGAS-STING pathway in colitis-associated colorectal cancer (CAC). The studies examined the effects of PARP-1 inhibition, STING signaling, and CDK4/6 inhibition on inflammation-driven tumorigenesis using AOM/DSS models. Each study highlighted distinct mechanisms by which these pathways regulate inflammation, immune responses, and tumor progression, offering insights into potential therapeutic interventions for CAC.

Study/Reference	Key Findings	Mechanisms	Therapeutic Insights
Ghonim et al. (2023)/ [75]	Partial inhibition of PARP-1 (via heterozygosity or low-dose olaparib) reduced tumor burden, colitis severity, and systemic inflammation; full inhibition less effective.	Modulated MDSC function; reduced TNF-α, MCP-1; enhanced mucosal integrity. Excess inhibition impaired immune homeostasis.	Partial PARP-1 inhibition shows robust protective effect against CAC; excessive inhibition may be counterproductive.
Ahn et al. (2017)/ [60]	STING-deficient mice had increased polyps, dysbiosis, and disrupted cytokine balance. Antibiotics reduced polyp load.	STING regulated IL-10, IL-1β, and IL-18 via MyD88-dependent pathways; microbial CDNs were dominant STING activators over self-DNA.	STING controls tumor-promoting inflammation; microbial modulation and STING targeting may suppress CAC progression.
Yang et al. (2023)/ [76]	Palbociclib reduced inflammation, epithelial damage, and tumor load in AOM/DSS mice; suppressed pro-inflammatory cytokines (Ifnb1, Il6, Il1b)	Inhibited STING activation by interfering with its dimerization and signaling cascade.	Palbociclib modulates STING to curb inflammation-driven CRC; further studies needed to confirm clinical potential.

**Table 2 ijms-26-04979-t002:** Summarizes the findings from studies on the role of the STING pathway in colitis-associated colorectal cancer (CAC). The table highlights key insights into STING activation at different stages of CAC progression, its mechanisms in modulating inflammation and immune responses, and the therapeutic potential of targeting the STING pathway. These studies reveal the dual role of STING in promoting inflammation-driven tumorigenesis and supporting anti-tumor immunity, emphasizing the importance of timing and context in developing STING-targeted therapies.

Study/Reference	Key Findings	Mechanisms	Therapeutic Insights
Cai et al. (2023)/ [58]	STING activation in early CAC exacerbates inflammation; myeloid-specific STING deletion reduces tumor formation. In advanced tumors, STING is essential for anti-tumor immunity.	STING promotes inflammation and tumor initiation via TFAM-mtDNA activation but supports CD8^+^ T cell responses in advanced disease.	Timing of STING-targeted therapy is critical. Early inhibition may prevent CAC, while later activation enhances anti-tumor responses.
Vasudevan et al. (2023)/ [79]	STING interacts with Syk to induce pyroptosis via GSDMD cleavage. STING deficiency promotes tumor growth and suppresses anti-tumor cytokine production.	STING–Syk interaction triggers pyroptosis and enhances type-I IFN production via TBK1-IRF3; regulates tumor cytokine balance.	STING agonists like DMXAA may enhance pyroptosis and immune responses. STING–Syk axis is a potential therapeutic target in CAC.

## Data Availability

All data are contained within the article.

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
