# Peer review of "Inflammatory Bowel Disease (IBD)-Associated Colorectal Cancer (CRC): Is cGAS-STING Pathway Targeting the Key to Chemoprevention?"

_ijms, 2025, doi:10.3390/ijms26114979_

Round 1
Reviewer 1 Report
Comments and Suggestions for Authors
There are several major problems with this review.
- The title of this review focuses on IBD-associated colorectal cancer, but there are too much IBD content in the main text. These two diseases are different.
- The specific mechanism diagrams of cGAS-STING in IBD-associated colorectal cancer needs to be shown in figures.
- The descriptions in columns 2, 3, 4 of Table 1 have similarities and should be distinguished. Similar situations also exist in other tables.
- Abbreviate when it first appear. There are many repeated abbreviations in the text, such as DAMPs, mtDNA, etc.
- The descriptions in lines 248-252 are repeated, but the references are different.
- The Part 4 lacks therapeutic strategies, such as inhibitors, natural products, etc.
- There are errors in the references, such as Reference 17.
- The Future Perspectives of the paper is not sufficient enough. More research suggestions should be provided for other researchers.
Author Response
Response to Reviewers
We would like to sincerely thank the reviewers for their insightful and constructive comments, which have significantly improved the quality, structure, and clarity of our manuscript. We truly appreciate the time and effort invested in reviewing our work.
As correctly pointed out by Reviewer 1, our original draft contained a disproportionately large section on inflammatory bowel disease (IBD), while the part discussing colorectal cancer associated with IBD—the primary focus of our review—was underdeveloped. This imbalance diverted attention from the core message of our manuscript. We agree that this structural issue could mislead the reader regarding the review’s primary objective. Moreover, the absence of a dedicated subsection on therapeutic strategies was an important oversight. In response, we have carefully restructured the manuscript: trimmed the IBD sections, enriched the CAC discussion (especially Section 4), and added a new subsection (4.4) titled “Emerging Therapeutic Approaches Targeting the cGAS-STING Pathway in CAC” to specifically address preclinical and translational therapeutic strategies. This paragraph synthesizes data from animal models, describes drug development challenges, and discusses how these strategies may influence future clinical practice.
We address each reviewer comment point by point below.
Reviewer 1
Comment 1: “The title of this review focuses on IBD-associated colorectal cancer, but there are too much IBD content in the main text. These two diseases are different.”
Response: We fully agree. The IBD section was disproportionately long and diverted attention from CAC. We have now extensively trimmed Sections 3.1–3.3, synthesized overlapping studies, and focused Section 4 more clearly on CAC-related mechanisms. The revised structure now aligns the manuscript with the title and main objectives.
Comment 2: “The specific mechanism diagrams of cGAS-STING in IBD-associated colorectal cancer need to be shown in figures.”
Response: Thank you for this valuable suggestion. We have added a new figure (now Figure 3) illustrating the dual role of cGAS-STING signaling in CAC pathogenesis, encompassing inflammation-driven carcinogenesis and anti-tumor immunity.
Comment 3: “The descriptions in columns 2, 3, 4 of Table 1 have similarities and should be distinguished. Similar situations also exist in other tables.”
Response: We revised Table 1 entirely to avoid redundancy and improve clarity. We also restructured Table 2 with a cleaner, more comparative format, now clearly distinguishing key findings, mechanisms, and therapeutic implications.
Comment 4: “Abbreviate when it first appear. There are many repeated abbreviations in the text, such as DAMPs, mtDNA, etc.”
Response: We reviewed and standardized all abbreviations throughout the manuscript, ensuring that each abbreviation is spelled out upon first use and subsequently abbreviated consistently.
Comment 5: “The descriptions in lines 248-252 are repeated, but the references are different.”
Response: We thank the reviewer for noticing this oversight. The repeated content has been consolidated and clarified to avoid redundancy.
Comment 6: “The Part 4 lacks therapeutic strategies, such as inhibitors, natural products, etc.”
Response: This was a critical point. We added Section 4.4 titled “Emerging Therapeutic Approaches Targeting the cGAS-STING Pathway in CAC,” which discusses STING agonists/antagonists, drug delivery limitations, cytokine-related side effects, and natural product modulators.
Comment 7: “There are errors in the references, such as Reference 17.”
Response: All references have been thoroughly reviewed, updated, and corrected where necessary.
Comment 8: “The Future Perspectives of the paper is not sufficient enough. More research suggestions should be provided for other researchers.”
Response: We rewrote the Conclusion and Future Perspectives as a unified section, highlighting unresolved questions, translational gaps, and proposing directions for future research, especially in the context of personalized therapy and surveillance strategies.
Reviewer 2 Report
Comments and Suggestions for Authors
This narrative review surveys the rapidly expanding evidence that the cGAS-STING DNA-sensing axis is a molecular pathway linking chronic intestinal inflammation to colitis-associated colorectal cancer (CAC). It synthesises data from murine colitis/CAC models, epithelial and immune-cell mechanistic studies and early translational work with STING agonists/antagonists. The subject is clinically relevant because conventional chemopreventive options (mesalazine, immunomodulators) leave unmet need in Crohn’s disease and high-risk ulcerative colitis subgroups. Strengths are its didactic figures and a comprehensive bibliography. However, the manuscript currently leans toward descriptive narrative rather than critical synthesis; it lacks a formal literature-search section, is near an upper word count limit, and sometimes overstates pre-clinical findings. Substantial revision is therefore required before it can meet IJMS quality standards for review articles.
Major comments:
- Although this is a narrative rather than a systematic review, a Methods section could be added in which the search metodology is described. Currently, there is no clear description of how the 87 papers were selected for this review.
- Since similar reviews already exist, for example see PMID: 37781354, PMID: 33833439, PMID: 38762923, the unique contribution of this review should be stated, ie. a paragraph should be added which compares earlier reviews and puts this review into perspective, highlighting its angle.
- Although there are no current clinical trials for STING agonists for IBD or CAC (which sould be mentioned somewhere), it would be welcome to further discuss translational issues such as: a. drug classes currently tested in humans, b. potential biomarkers, c. Delivery and safety obstacles; for example systemic STING agonists cause cytokine spikes (PMID: 39421752). Furthermore, murine-active DMXAA failed in humans (PMID: 23585680).
- Figures and tables. Table 1 mostly duplicates the text, should be made more concise. Same for Table 2 (text duplication). Each table should not break the text describing one particular study (which should be trimmed to a one-line bullet).
- Further major papers which could be added for their translational-clinical importance: PMID: 39448330 (State-of-the-art review of cGAS-STING in experimental colitis and CAC), PMID: 39611480 (Provides 2024 data on cGAS-STING across GI cancers, including CRC and CAC models).
Minor comments
- Define IEC, AHR and R-loop at first mention.
- Provide Conflict of Interest, Funding, Author contributions.
Row 471 STING (capitalize)
Row 476 Enteric
My recommendation is major revision.
Author Response
Response to Reviewers
We would like to sincerely thank the reviewers for their insightful and constructive comments, which have significantly improved the quality, structure, and clarity of our manuscript. We truly appreciate the time and effort invested in reviewing our work.
As correctly pointed out by Reviewer 1, our original draft contained a disproportionately large section on inflammatory bowel disease (IBD), while the part discussing colorectal cancer associated with IBD—the primary focus of our review—was underdeveloped. This imbalance diverted attention from the core message of our manuscript. We agree that this structural issue could mislead the reader regarding the review’s primary objective. Moreover, the absence of a dedicated subsection on therapeutic strategies was an important oversight. In response, we have carefully restructured the manuscript: trimmed the IBD sections, enriched the CAC discussion (especially Section 4), and added a new subsection (4.4) titled “Emerging Therapeutic Approaches Targeting the cGAS-STING Pathway in CAC” to specifically address preclinical and translational therapeutic strategies. This paragraph synthesizes data from animal models, describes drug development challenges, and discusses how these strategies may influence future clinical practice.
We address each reviewer comment point by point below.
Reviewer 2
Comment 1: “Although this is a narrative rather than a systematic review, a Methods section could be added in which the search methodology is described.”
Response: We have added a short methodology paragraph under the Introduction explaining the literature search strategy and inclusion scope.
Comment 2: “Since similar reviews already exist, for example see PMID: 37781354, PMID: 33833439, PMID: 38762923, the unique contribution of this review should be stated.”
Response: We now include a paragraph in the Introduction comparing our work to previous reviews, highlighting that ours uniquely integrates epithelial, immune, and translational perspectives with a focus on CAC-specific mechanisms.
Comment 3: “Although there are no current clinical trials for STING agonists for IBD or CAC... discuss translational issues such as drug classes currently tested in humans, potential biomarkers, delivery and safety obstacles.”
Response: This was addressed in the new Section 4.4, which provides a balanced discussion on translational barriers, including cytokine spikes, delivery system limitations, and clinical relevance.
Comment 4: “Figures and tables. Table 1 mostly duplicates the text, should be made more concise. Same for Table 2.”
Response: Both tables have been significantly revised to reduce redundancy, clarify distinctions, and enhance readability.
Comment 5: “Further major papers could be added for their translational-clinical importance.”
Response: We added new references, including PMIDs 39448330 and 39611480, in the therapeutic discussion and revised the bibliography accordingly.
Minor comments:
- IEC, AHR, R-loop are now defined at first mention.
- Conflict of Interest, Funding, and Author Contributions sections have been added.
- Capitalization issues such as “STING” and spelling corrections like “enteric” have been resolved.
Reviewer 3 Report
Comments and Suggestions for Authors
The manuscript presents an impressively comprehensive overview of the cGAS‑STING pathway in inflammatory‑bowel‑disease‑associated colorectal cancer. Its breadth of primary‑literature coverage will interest readers; however, a number of substantive issues currently should be considered.
- The current draft contains extensive repetition of the same mechanistic anecdotes under several headings, which risks obscuring the broader conclusions the review seeks to convey. Moreover, the manuscript summarizes individual papers at length yet offers little synthesis or discussion of conflicting data. Instead of weaving individual studies into a cohesive narrative, the text offers prolonged, paper‑by‑paper summaries that read like annotated abstracts. This approach obscures the bigger picture and forces the reader to infer connections that the authors should be making explicitly. Summarise individual experiments in one or two sentences, then devote the paragraph to what those studies collectively teach us.
- No search strategy, inclusion/exclusion criteria, or quality‐assessment approach is described.
- Tables are dense paragraph‑style blocks rather than concise comparative summaries. A concise matrix mapping each key paper to “cell type, stimulus, read‑out, net effect (protective vs. pathogenic)” would let the reader grasp patterns at a glance.
- Replace the legacy statistics with the most recent population‑based to ensure the review reflects contemporary clinical reality.
- Spell out the full term followed by the abbreviation in parentheses and thereafter use the abbreviation alone.
Author Response
Response to Reviewers
We would like to sincerely thank the reviewers for their insightful and constructive comments, which have significantly improved the quality, structure, and clarity of our manuscript. We truly appreciate the time and effort invested in reviewing our work.
As correctly pointed out by Reviewer 1, our original draft contained a disproportionately large section on inflammatory bowel disease (IBD), while the part discussing colorectal cancer associated with IBD—the primary focus of our review—was underdeveloped. This imbalance diverted attention from the core message of our manuscript. We agree that this structural issue could mislead the reader regarding the review’s primary objective. Moreover, the absence of a dedicated subsection on therapeutic strategies was an important oversight. In response, we have carefully restructured the manuscript: trimmed the IBD sections, enriched the CAC discussion (especially Section 4), and added a new subsection (4.4) titled “Emerging Therapeutic Approaches Targeting the cGAS-STING Pathway in CAC” to specifically address preclinical and translational therapeutic strategies. This paragraph synthesizes data from animal models, describes drug development challenges, and discusses how these strategies may influence future clinical practice.
We address each reviewer comment point by point below.
Reviewer 3
Comment 1: “The current draft contains extensive repetition… manuscript summarizes individual papers at length yet offers little synthesis.”
Response: We revised all mechanistic sections (Sections 3.1–3.3) to reduce repetition and provide integrative analysis instead of sequential summaries. Studies are now grouped by cell type and theme, and overarching conclusions are drawn explicitly.
Comment 2: “No search strategy, inclusion/exclusion criteria, or quality‐assessment approach is described.”
Response: A search strategy is now included under the Introduction.
Comment 3: “Tables are dense paragraph‑style blocks… A concise matrix mapping each key paper to ‘cell type, stimulus, read‑out, net effect’ would let the reader grasp patterns at a glance.”
Response: We restructured the tables into matrix format with cleaner headers (e.g., Study, Cell Type, Stimulus, Mechanism, Net Effect), allowing for easier interpretation.
Comment 4: “Replace the legacy statistics with the most recent population‑based data.”
Response: We updated the epidemiological data using the 2020 Lancet comparative study (Ola Olén et al., PMID: 31929014), and now reference current incidence and mortality trends.
Comment 5: “Spell out full term followed by abbreviation at first use.”
Response: This has been implemented throughout the text.
We believe that these revisions have strengthened the manuscript and brought it in line with the journal’s high standards. We are grateful for your consideration and look forward to your feedback.
Round 2
Reviewer 1 Report
Comments and Suggestions for Authors
This new manuscript version can be accepted now.
Author Response
We sincerely thank the reviewer for their constructive and thoughtful comments, which have helped us further improve the clarity and scientific quality of our manuscript.
Reviewer 2 Report
Comments and Suggestions for Authors
The authors have now improved the manuscript significantly, and have added the necessary explanations.
There are some further suggestions (in line with the previous comments):
- Some specific (candidate) biomarkers should be addressed briefly (now mentioned as “genetic, microbial, and immunologic biomarkers” at 802 and 803), such as IFI16, MB21D1 (cGAS), TMEM173 (STING) and TBK1). See: PMID: 38831059
- It should be mentioned explicity that no STING‑modulator trials have yet been registered for IBD or CAC. Several (mentioned) human‑specific STING agonists (RR‑CDA/MIW815, SNX281, diABZI) are in phase I/II oncology trials, however no interventional studies have yet been registered for inflammatory‑bowel‑disease or CAC indications.
Minor English copy‑editing.
Author Response
We sincerely thank the reviewer for their constructive and thoughtful comments, which have helped us further improve the clarity and scientific quality of our manuscript.
Comment:
“Some specific (candidate) biomarkers should be addressed briefly (now mentioned as “genetic, microbial, and immunologic biomarkers” at 802 and 803), such as IFI16, MB21D1 (cGAS), TMEM173 (STING) and TBK1). See: PMID: 38831059.”
Response:
Thank you for this valuable suggestion. We have now revised the relevant section to include a concise summary of these candidate biomarkers based on the findings by Wang et al. (Sci Rep, 2024; PMID: 38831059). Specifically, we now highlight IFI16, MB21D1 (cGAS), TMEM173 (STING), and TBK1 as key cGAS–STING-related genes upregulated in ulcerative colitis, with IFI16 showing particular promise as a diagnostic and predictive biomarker for anti-TNF treatment response.
Comment:
“It should be mentioned explicitly that no STING‑modulator trials have yet been registered for IBD or CAC. Several (mentioned) human‑specific STING agonists (RR‑CDA/MIW815, SNX281, diABZI) are in phase I/II oncology trials, however no interventional studies have yet been registered for inflammatory‑bowel‑disease or CAC indications.”
Response:
We thank the reviewer for this important observation. In response, we have revised Section 4.4 of the manuscript to clearly state that, while several STING agonists such as MIW815 (RR-CDA), SNX281, and diABZI are currently undergoing clinical evaluation in oncology trials, no interventional studies involving STING-targeting agents have yet been registered for inflammatory bowel disease (IBD) or colitis-associated colorectal cancer (CAC). This addition highlights the existing translational gap and strengthens the relevance of our discussion on future therapeutic directions.
Reviewer 3 Report
Comments and Suggestions for Authors
I think the authors have adequately responded to the comments presented.
Author Response

(The authors gave the same response as above.)
